# *nnUNet meets pathology*: bridging the gap for application to whole-slide images and computational biomarkers

**Joey Spronck**[1]                                                    JOEY.SPRONCK@RADBOUDUMC.NL
**Thijs Gelton**[1]                                                    THIJS.GELTHON@RADBOUDUMC.NL
**Leander van Eekelen**[1]                              LEANDER.VANEEKELEN@RADBOUDUMC.NL
**Joep Bogaerts**[1]                                        JOEP.BOGAERTS@RADBOUDUMC.NL
**Leslie Tessier**[1]                                          LESLIE.TESSIER@RADBOUDUMC.NL
**Mart van Rijthoven**[1]                              MART.VANRIJTHOVEN@RADBOUDUMC.NL
**Lieke van der Woude**[1]                          LIEKE.VANDERWOUDE@RADBOUDUMC.NL
**Michel van den Heuvel**[2]                  MICHEL.VANDENHEUVEL@RADBOUDUMC.NL
**Willemijn Theelen**[3]                                              W.THEELEN@NKI.NL
**Jeroen van der Laak**[1]                            JEROEN.VANDERLAAK@RADBOUDUMC.NL
**Francesco Ciompi**[1]                              FRANCESCO.CIOMPI@RADBOUDUMC.NL

[1] *Department of Pathology, Radboud University Medical Center, Nijmegen, Netherlands*

[2] *Respiratory Diseases Department, Radboud University Medical Center, Nijmegen, Netherlands*

[3] *Thoracic Oncology Department, Netherlands Cancer Institute, Amsterdam, Netherlands*

**Editors:** Accepted for publication at MIDL 2023

## Abstract

Image segmentation is at the core of many tasks in medical imaging, including the engineering of computational biomarkers. While the self-configuring nnUNet framework for image segmentation tasks completely shifted the state-of-the-art in the radiology field, it has never been applied to the field of histopathology, likely due to inherent limitations that nnUNet has when applied to the pathology domain "off-the-shelf". We introduce nnUNet for pathology, built upon the original nnUNet, and engineered domain-specific solutions to bridge the gap between radiology and pathology. Our framework includes critical hyperparameter adjustments and pathology-specific color augmentations, as well as an essential whole-slide image inference pipeline. We developed and validated our approach on non-small cell lung cancer data, showing the effectiveness of nnUNet for pathology over default nnUNet settings, and achieved the first position on the experimental segmentation task of the TIGER challenge on breast cancer data when using our pipeline "out-of-the-box". We also introduce a novel inference uncertainty approach, which proved helpful for the quantification of the tumor-infiltrating lymphocytes biomarker in non-small cell lung cancer biopsies of patients treated with immunotherapy. We coded our framework as a workflow-friendly segmentation tool and made it publicly available.[1]

**Keywords:** nnUNet, computational pathology, segmentation, TIL biomarker, uncertainty.

## 1. Introduction

Computer vision tasks such as semantic segmentation and object detection are the foundation of automated image analysis and understanding. In medical imaging, these tasks function at the core of computer-aided detection and diagnosis systems (Gao et al., 2019),

---

1. https://github.com/DIAGNijmegen/nnUNet-for-pathology

and the engineering of computational biomarkers in both radiology (Parekh et al., 2020) and histopathology images (Kulkarni et al., 2020). The advent of deep learning (LeCun et al., 2015) facilitated a substantial breakthrough in the development of accurate detection and segmentation models, powering systems that achieve and even surpass human performance on certain medical tasks (Esteva et al., 2017). In semantic image segmentation, the UNet architecture (Ronneberger et al., 2015) has increasingly been adopted and has become the state-of-the-art (SOTA) approach. However, hyperparameter tuning of UNet architectures is a crucial, non-trivial, and time-consuming endeavor for training highly specialized medical imaging solutions.

In recent years, AutoML approaches have been proposed to address this problem (He et al., 2021). In the field of radiology, Isensee et al. (2020) introduced the self-configuring nnUNet framework, which utilizes well-established internal fingerprints, fixed parameters, empirical decisions, and a cross-validation strategy to determine optimal hyperparameters to train a UNet model. This self-configuring approach enabled nnUNet to surpass the SOTA performance in many radiology applications (Antonelli et al., 2022). While transformer-based segmentation approaches are increasingly being adopted (Chen et al., 2022), UNet-based models with manually tweaked hyperparameters still power SOTA applications in the computational pathology domain (Bulten et al., 2020; Ho et al., 2021). Surprisingly, the challenge-winning self-configuring nnUNet framework has never been adapted to suit the requirements for application to the pathology domain.

Despite the general purpose nature of nnUNet, several aspects of pathology data are intrinsically different from radiology. For example, loading entire gigapixel histopathology whole-slide images (WSIs) poses memory constraints during training and inference, resorting to customized dataset preparation and sliding window techniques that are not included in nnUNet's default framework. Additionally, multi-class semantic segmentation in histopathology involves a large variety of patterns, cell types, and tissue morphologies, which may not be fully considered in nnUNet's radiology-oriented default configurations.

To accommodate for the high morphological variety, segmentation models for pathology tasks typically use Batch Normalization (BN) with an appropriate batch size, in contrast with nnUNet's preference for large patch sizes and smaller batch sizes, in combination with Instance normalization (IN). Additionally, capturing the full spectrum of tissue and cellular varieties in the training data is difficult, and deep learning models often make incorrect, overconfident predictions for out-of-training-distribution data. Extracting a model's uncertainty can therefore be useful in identifying poorly segmented regions when applied to unseen data (Lakshminarayanan et al., 2016). Finally, pathology data often exhibits variations in slide staining, even within the same lab on different days. To address this, pathology-specific data augmentation techniques are commonly used (Tellez et al., 2019).

In this paper, we build upon the existing nnUNet framework and, to the best of our knowledge, propose the first application of nnUNet to digital pathology data. We identify and address pathology-specific workflow-related limitations of the framework, critically assess nnUNet's model configurations on a multi-class tissue segmentation task on hematoxylin and eosin (H&E)-stained non-small cell lung cancer (NSCLC) WSIs, and introduce a lightweight uncertainty extraction method to identify likely-incorrectly segmented regions.

To prove the versatility and robustness of the entire nnUNet framework for pathology data, we assess the proposed configuration's training and WSI-inference performance on the

most recent public pathology benchmark for multi-class segmentation; the TIGER challenge [2]. This challenge involves a different organ and segmentation task and therefore serves as a perfect performance readout of our work against optimized SOTA approaches.

Finally, we showcase the use of the proposed nnUNet framework in biomarker development on a tumor-infiltrating lymphocytes (TILs) quantification task in a cohort of metastatic NSCLC cases treated with immunotherapy (Theelen et al., 2019). While initially intra-tumoral TILs ($_i$TILs) were deemed most relevant, (Salgado et al., 2015) proposed visual estimation in the stromal regions of the tumor ($_s$TILs), while excluding necrotic areas. Since this approach depends on the identification of multiple tissue types, it was adopted as a suitable proof of concept segmentation task for nnUNet's use in biomarker development.

## 2. Methods

We first describe the data used during training, validation, and biomarker development. Then, we introduce nnUNet for pathology, where we focus on added technical and methodological contributions to the framework, and go over the experiments carried out in an ablation study to verify that the proposed changes improve segmentation performance over the default nnUNet framework. Finally, we detail how the proposed nnUNet framework was evaluated on the TIGER challenge, and explain the steps taken for TIL biomarker development.

### 2.1. Materials

**Train and test data.** For training, we used a set of 30 H&E stained WSI of NSCLC tissue samples (21 surgical resections, 9 biopsies) from Radboud University Medical Center (Nijmegen, Netherlands). Cases were selected to represent a balanced mix of adenocarcinoma (AD) and squamous cell carcinoma (SC) subtypes, tumor growth patterns, and morphologies. For each WSI, approximately five regions of interest (ROI) of $2 \times 2$ $mm$ were selected by a pathologist and manually annotated densely to cover 13 morphological patterns in the tissue (Fig. 1). Using the same selection and annotation procedure, we obtained an external test set containing 14 cases (7 AD, 7 SC) from the publicly available TCGA-LUAD and TCGA-LUSC datasets (respectively containing AD and SC cases). Additionally, test cases were sought out to contain varying stain color characteristics, to assess the generalizability of trained nnUNet models to staining variations across different medical centers. In total, 163 training ROIs and 55 test ROIs were annotated; an example of the data is depicted in Fig. 1.

**TIGER challenge data.** The challenge's publicly available training data contains 195 WSI from various medical centers with a total of 286 annotated ROIs. Each ROI contains up to 7 classes and may contain unannotated regions that should be excluded from loss calculation. The experimental test used for the SOTA comparison of our proposed nnUNet framework contained 26 WSI with 130 ROIs. The final test set was not considered in this study because it was not accessible at the time of validating our method.

**PEMBRO-RT trial data.** The predictive value of TILs was assessed on a cohort of NSCLC treated with immunotherapy (Pembrolizumab) from PEMBRO-RT (Theelen et al., 2019), a phase-2 clinical trial led by the Dutch Cancer Institute (NKI) in the Netherlands. This cohort consists of 68 tumor biopsies from patients with metastatic NSCLC, containing

---

2. https://tiger.grand-challenge.org/

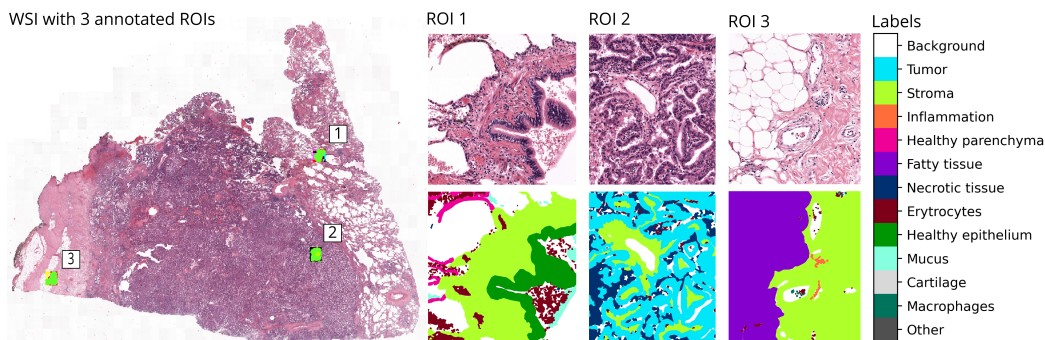

**Figure 1:** Visual example of a WSI with 3 annotated ROIs; labels legend indicates names of annotated classes and their corresponding colors.

25 biopsies from the lung and 43 from other metastatic sites. Since the train and test data in this study solely contains lung tissue, the selection was restricted to the 25 lung tissue biopsies. Furthermore, two cases were excluded due to insufficient biopsy quality, assessed by a resident pathologist, resulting in n=23 cases. As a readout for immunotherapy treatment response, we defined response as stable disease (as defined by the RECIST1.1 guidelines (Eisenhauer et al., 2009)) or better at 12 weeks after treatment.

## 2.2. nnUNet for pathology

**From whole-slide images to nnUNet and back.** We developed a custom dataset preparation pipeline for the nnUNet framework that is compatible with pathology data. Our pipeline uses the open-source WholeSlideData Python library[3] to fast and efficiently extract ROIs and corresponding annotations from the WSIs on a set spacing of 0.5 $\mu$m/px. These extracted ROIs and annotation masks are then converted to the required nifti file format for compatibility with nnUNet's pre-processing pipeline. We ensured similar class distributions in the train and validation splits instead of using a random 5-fold cross-validation split.

Additionally, we developed a WSI inference approach as an essential feature for nnUNet for pathology. The pipeline (1) accepts a WSI and its tissue/background mask (Bándi et al., 2019), (2) samples patches from the WSI, using a sliding window on patches containing tissue, (3) applies nnUNet's default sliding window approach with Gaussian weighted half overlap within the sampled patches, (4) only writes the fully overlapped region of the sampled patches in a pyramidal TIFF file structure, and (5) outputs the model output mask, and uncertainty mask (further described in the 'Inference uncertainty extraction' section). Further details of the WSI inference pipeline are discussed in the Appendix.

**WSI segmentation with nnUNet.** Evaluated configurations of the nnUNet model are: types of input pre-processing, feature normalization methods, batch size and patch size configurations, and the use of pathology-specific color augmentations. The performance of the evaluated changes was measured using the f1-score, which combines the precision and recall into a single metric by taking the harmonic mean.

---

3. https://github.com/DIAGNijmegen/pathology-whole-slide-data

*Input pre-processing.* In contrast with radiology image pixels representing signal intensities, pathology image pixels represent color in RGB space. The default nnUNet preprocessing method, z-score normalization, normalizes each image channel separately with a mean of 0 and a standard deviation of 1. However, this method may result in color shifts in pathology images' RGB channels. To mitigate this issue, we evaluated the performance of 0-1 scaling, which scales 8-bit images with values ranging from 0 to 255 to a new range of 0-1 without causing color shifts.

*Feature normalization & patch and batch size.* We noticed experimentally that the default settings of nnUNet can lead to a 'label switching' patch effect (Fig. 5 A). Default nnUNet strongly favors large patch sizes over large batch sizes, which supports the use of IN that, in contrast with BN, works well with extremely small batch sizes. We evaluated the segmentation performance and the impact on this patch effect of both normalization methods on a model with patch size 1024×1024 px (ps1024) and batch size 2 (bs2) (configured by default nnUNet), and patch size 512×512 px (ps512) and batch size 8 (bs8).

*Data augmentation.* We assessed the use of Hematoxylin-Eosin-DAB (HED) and Hue-Saturation-Value (HSV) augmentation for nnUNet. HED augmentation decomposes an RGB image into the staining intensities of the individual hematoxylin, eosin, and DAB staining, augments their intensities, and converts them back to RGB. Similarly, HSV augmentation first converts the RGB images to HSV space, augments the images' hue, saturation, and value, and converts the augmented image back to RGB space. We evaluated the addition of HED, and the sequential addition of HED and HSV, as the first color-augmenting transformations in the default data augmentation pipeline. From the HED and HSV augmentation settings proposed in Tellez et al. (2019), we used the HED-light setting, and an adjusted HSV-light setting, with additional value augmentation in the same intensity range as the hue and saturation.

Next to augmentations during training, nnUNet also encompasses test time augmentation (TTA), which mirrors the input image along the two axes during inference and reconstructs and averages the predictions on the augmented images into a final output. The effect of TTA was evaluated on all before-mentioned nnUNet variants.

**Inference uncertainty extraction.** By default, nnUNet trains five models through a 5-fold cross-validation approach. These models enable the extraction of pixel uncertainty, derived from the internal disagreement among the 5 folds, where greater disagreement corresponds to more uncertain pixel predictions. To calculate a fold's disagreement on a pixel, we use the soft cross-entropy loss over the fold's pixel $log(softmax)$ values and the mean pixel $log(softmax)$ values of the 5 folds. The final uncertainty value per pixel is obtained by averaging the disagreements from the 5 folds, following a method similar to that proposed by Lakshminarayanan et al. (2016). Given that uncertain predictions tend to be more incorrect, we use the uncertainty value as a measure of incorrectness. We examine the uncertainty value distribution for each predicted output class on the entire test set, splitting True Positive (TP) and False Positive (FP) predictions' uncertainty values for each class. We established a cutoff point in the uncertainty values for each class individually, using the ROC curves' optimal operation points to distinguish TP from FP predictions based on the uncertainty value.

## 2.3. TIL biomarker development

As a proof of concept biomarker development application for nnUNet, we investigate different TIL densities on each of the 23 lung biopsies of metastatic NSCLC from the PEMBRO-RT trial (Theelen et al., 2019). The HoVerNet algorithm, a nuclei segmentation and classification model which was trained on multiple organ tissues (Graham et al., 2019), was used to detect inflammatory cells as a proxy for TILs (we validated its use as TIL detector using manual annotations, see Appendix Fig. 6). TIL densities were computed within the whole tissue ($_{tissue}$TILs), nnUNet's stromal sections (for $_s$TILs), and nnUNet's tumor segments (for $_i$TILs). Here, stromal segments are defined as the union of nnUNet's 'stroma' and (stromal) 'inflammation' class. Additionally, we investigate the use of our proposed uncertainty filtering approach to exclude likely-FP regions from the nnUNet's tumoral and stromal regions. The derived TIL densities were used to stratify responders using ROC analysis. Welsh test statistics were used to evaluate the significance of the TIL density distributions from responders and non-responders originating from different populations.

## 2.4. TIGER challenge segmentation

We adapted the proposed and default configurations to suit the labels of the TIGER dataset, trained the 5-folds, and subsequently applied them through our WSI inference pipeline on the experimental test set on the Grand Challenge platform, returning the final f1 scores of the proposed and default configurations.

## 3. Results

**nnUNet for pathology**   Out-of-the-box application of nnUNet to pathology data showed promising quantitative results on patch segmentation level. However, we identified a 'label-switching' patch effect on adjacent or overlapping patches (Appendix Fig. 5 A) inherent to nnUNet's default configuration, which is detrimental to WSI inference performance. By conducting an ablation study, we identified several critical changes and beneficial additions that are necessary to improve nnUNet's default configurations for its application to pathology data. An overview of the evaluated nnUNet configurations is shown in Table 1.

**Table 1:** nnUNet experiments table

| | nnUNet configuration | | | | F1 score (F1 score without TTA) | | |
|---|---|---|---|---|---|---|---|
| | input scale | feature norm | batch/patch size | DA | Micro avg | Macro avg | Weighted avg |
| **A** | z-score | IN | 2/1024 | default | 0.697 (0.692) | 0.378 (0.374) | 0.655 (0.649) |
| **B** | 0-1 | IN | 2/1024 | default | 0.695 (0.691) | 0.382 (0.379) | 0.652 (0.648) |
| **C** | 0-1 | IN | 8/512 | default | 0.728 (0.728) | 0.406 (0.408) | 0.690 (0.690) |
| **D** | 0-1 | BN | 2/1024 | default | 0.752 (0.750) | 0.486 (0.485) | 0.725 (0.723) |
| **E** | 0-1 | BN | 8/512 | default | 0.765 (0.754) | 0.511 (0.510) | 0.735 (0.733) |
| **F** | 0-1 | BN | 8/512 | +HED | **0.768 (0.766)** | **0.522 (0.520)** | **0.748 (0.746)** |
| **G** | 0-1 | BN | 8/512 | +HED,HSV | 0.766 (0.764) | **0.522 (0.520)** | 0.746 (0.743) |

*Input scaling.* nnUNet's default z-score normalization can introduce color shifts in adjacent or overlapping patches, which may contribute to the label-switching patch effect. However, using 0-1 scaling did not resolve the issue (Appendix Fig. 5 B). Although the performance of both methods on patch segmentation level is comparable (Table 1 A and B), 0-1 scaling is preferred since it avoids color channel disturbance and is required for HED and HSV augmentation.

*Feature normalization & batch size, patch size.* The performance of the evaluated models with different feature normalization approaches, and batch and patch sizes are depicted in Table 1 (B, C, D, and E). Both replacing IN with BN and using bs8 ps512 instead of bs2 ps1024 yielded superior results. Replacing the feature normalization approach yielded a major performance increase compared to changing the batch and patch size. A combination of both adjustments showed a synergistic performance increase on all computed f1 scores, with a macro averaged f1 score of 0.510 compared to 0.379 for both default settings. All evaluated variations from the default solved the identified patch effect (Appendix Fig. 5).

*Data augmentation.* Both HED and the combination of HED and HSV improved segmentation results compared to nnUNet's default data augmentations (1 E, F, G). Although very close to the configuration with HED and HSV augmentation, the addition of HED only (example augmentations in Appendix Fig. 10) yielded the highest overall performance in all averaged f1 scores and is thus proposed as the final nnUNet configuration for pathology. A WSI inference comparison example of nnUNet's default configuration versus our proposed configuration is shown in Fig. 2.

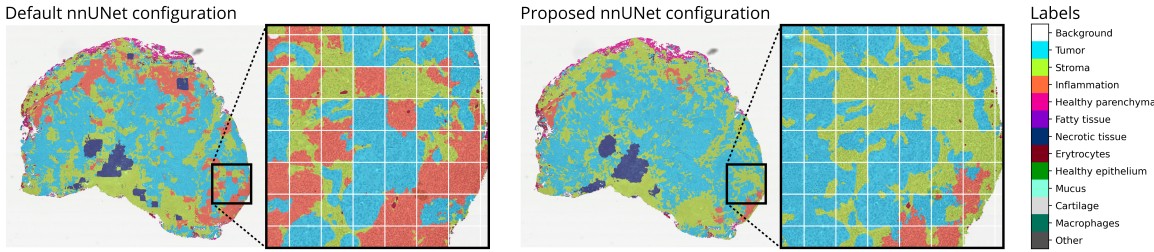

**Figure 2:** Comparison of WSI inference output between the default nnUNet configuration with the 'label switching' patch effect (left) and our proposed nnUNet configuration with improved accuracy and consistency (right), using the same inference pipeline with weighted half overlap. Our proposed model correctly identifies the tumor and stroma morphologies, while the default configuration exhibits the patch effect, incorrectly predicting different labels in adjacent patches with similar morphology, despite regularly predicting a single label within the patch (emphasized by the grid).

*Inference uncertainty extraction.* Appendix Fig. 7 demonstrates two visual examples of uncertainty extracted from the 5-fold disagreement, revealing that uncertain predictions are more likely to be incorrect. In Appendix Fig. 9, the complete result of our approach to differentiate TP from FP-predicted class pixels using the class' uncertainty values is depicted. The TP and FP predictions for each class exhibit clear differences in the uncertainty distributions, with TP predictions having mostly low uncertainty values and FP predictions having high uncertainty values. The distinction in uncertainty distributions was utilized to determine an optimal split point to distinguish likely-FP from likely-TP pixel predictions through ROC analysis. Fig. 3 A shows the uncertainty distributions and the defined cutoff value from the classes used for TIL biomarker development. Fig. 3 B provides a visual example of the effectiveness of uncertainty filtering in removing large incorrectly predicted tumor segments, which leads to a more accurate area for $i$TIL density calculation.

**TILs to predict immunotherapy response**  ROC curves on response stratification performance of our evaluated TIL density features are shown in Fig. 4. These results demonstrate that nnUNet-derived $s$TIL and $i$TIL features outperformed the overall tissue

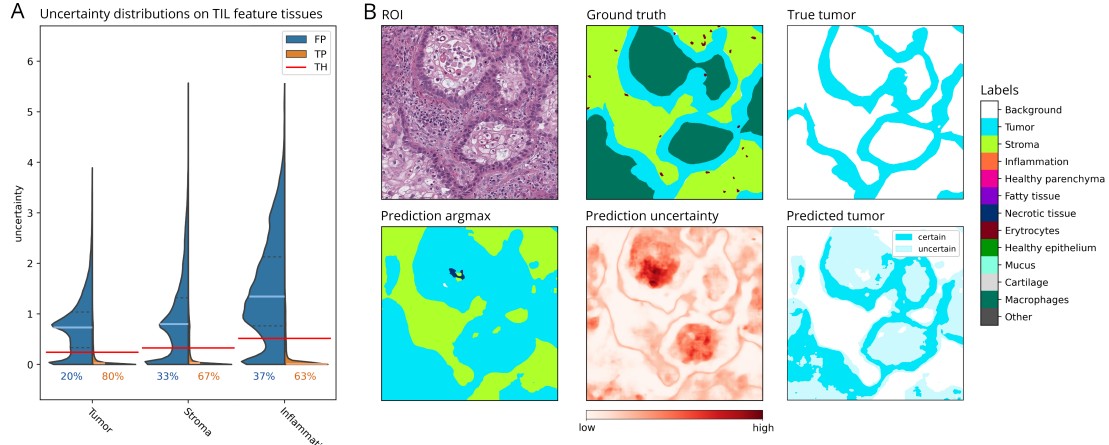

**Figure 3:** **A.** Pixel uncertainty distributions for $s$TIL and $i$TIL regions. Percentages depict the sizes of the distributions (TP: orange, FP: blue), with red thresholds (TH) indicating the optimal operating points for separating TP from FP. **B.** Example of uncertainty filtering. The model predicted foamy macrophages (out of training distribution) as tumor. We used the internal disagreement of the 5 folds to calculate pixel-level uncertainty values. Using the tumor TH, we separated likely-TP from likely-FP tumor pixels, where likely-TP pixels more closely resemble the true tumor mask.

TIL density feature, with AUCs of 0.843 and 0.799, respectively (Fig. 4 B and C). Furthermore, using our uncertainty approach to exclude likely-FP regions in the tumor and stromal segments improved the discriminatory power of both nnUNet-derived TIL density features, with AUCs of 0.876 and 0.826 for $s$TIL and $i$TIL, respectively. All TIL density features showed significant differences in Welsh test statistics (p-values $< 0.05$), but only the $_s$TIL density features remained significant after Bonferroni correction.

**TIGER challenge segmentation performance** On our first try, our proposed nnUNet for pathology framework with additional TTA ranked first on the segmentation task of the experimental leaderboard on detection and segmentation, with an f1 score of 0.8089, while the default configuration yielded an f1 score of 0.7741.

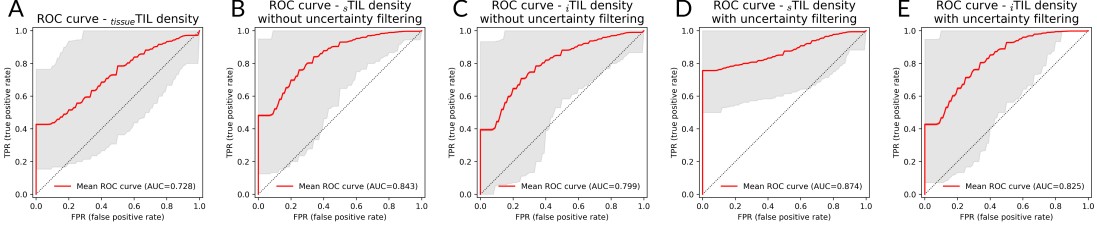

**Figure 4:** ROC curves with confidence interval and AUCs from different TIL density features, depicting that $s$TIL features and uncertainty filtering yield the best results.

## 4. Discussion

Our study demonstrates the potential of nnUNet in computational pathology by identifying and addressing its limitations in training and inference. The improvements proposed in this study, including the switching from IN to BN and increasing the batch size at the cost of

a reduction in patch size, are effective in addressing the high variability in cell morphology in histopathology WSI. Furthermore, by training on data from a single medical center and testing on external datasets, we validated the effectiveness of pathology-specific color augmentations in improving model robustness to different H&E stainings.

The nnUNet configuration encompassing 0-1 input scaling, BN, ps512, bs8, and additional HED augmentation yielded the best performance on all averaged f1 scores and defines our proposed nnUNet model configuration. The success of this proposed nnUNet model configuration and the entire WSI inference pipeline is further corroborated by the results on the TIGER challenge's segmentation task.

The development of the WSI inference pipeline represents a crucial contribution to the application of nnUNet in pathology. This essential tool is instrumental in bridging the practical gap that currently exists in utilizing nnUNet for pathology applications. Our successful resolution of the 'label switching' patch effect through proposed changes enabled accurate WSI inference performance, rendering nnUNet more appealing for use in pathology.

As a proof of concept for nnUNet's application in biomarker development, we successfully segmented tumor and stromal regions, yielding improved $_s$TIL and $_i$TIL density features compared to the overall $_{tissue}$TIL density. Our novel inference uncertainty implementation was supported by the improved AUC values achieved by filtering likely-FP tissue segments using pixel uncertainty cutoffs. Although being applied on a small cohort, the $_s$TILs demonstrated significant predictive value, as determined by Welsh test statistics, even after Bonferroni correction, showing the relevance of a high-quality segmentation as a base for the definition of biomarkers.

In addition to the improvements we have already made to nnUNet, there are further opportunities for enhancing the framework's performance in computational pathology. One such area is the optimization of the model configuration. While we manually adjusted the batch and patch size for our experiments, nnUNet has an automated configuration optimization feature that could be adjusted to meet specific requirements. Furthermore, the current sampling strategy and static dataset requirement may not be optimal for many pathology applications, such as cases with vast class imbalance or fully annotated whole slide images. To address these challenges, we plan to implement dynamic data loaders and remove the requirement for a static dataset to initialize the model configuration in our publicly available codebase, expanding the applicability of nnUNet in pathology.

## 5. Conclusion

Our proposed nnUNet for pathology framework has demonstrated its potential to contribute to the SOTA in the field of computational pathology. The results from our paper, including the first place on the TIGER challenge's (experimental) segmentation task, corroborate the effectiveness of our proposed configurations. Our approach provides an accurate and workflow-friendly segmentation tool, and we hope that the development of our WSI inference pipeline is welcomed by the computational pathology community, together with our novel inference uncertainty approach, and that can be adopted as a solid base for the development of segmentation tools. We also hope that the release of our code will encourage and facilitate further developments in the field of computational pathology.

## Acknowledgments

The authors would like to thank Dr. Fabian Isensee for his valuable assistance in this study and his help in identifying areas that could be improved, as well as the NKI-AVL Core Facility Molecular Pathology & Biobanking (CFMPB) for supplying NKI-AVL biobank material and/or lab support. This project was funded by a research grant from the Netherlands Organization for Scientific Research (NWO), project number 18388, and from the European Union's Horizon 2020 research and innovation program under grant agreement No 825292 (ExaMode, http://www.examode.eu/).

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

## Appendix A. WSI inference pipeline

We developed a WSI inference approach as an essential feature for nnUNet for pathology. At the core of the pipeline is a batch iterator from our WholeSlideData library, that enables users to sample patches from the data efficiently, fast, and easily. The Batch iterator loops patch-wise over the WSI, and writes the model's argmax output on the 5-fold's softmax values, and pixel-wise uncertainty map into 2 separate tiff files. The pipeline efficiently samples patches containing tissue by using a tissue/background mask (Bándi et al., 2019). It samples patches bigger than the model input size, making use of nnUNet's Gaussian sliding window approach, and only saves the inner part with full 4x overlap. To speed up the inference time, large parts of background are stripped of the patch (while preserving constant overlap) before inference. nnUNet's TTA is turned off with minimal/no performance drop.

## Appendix B. Label switching patch effect

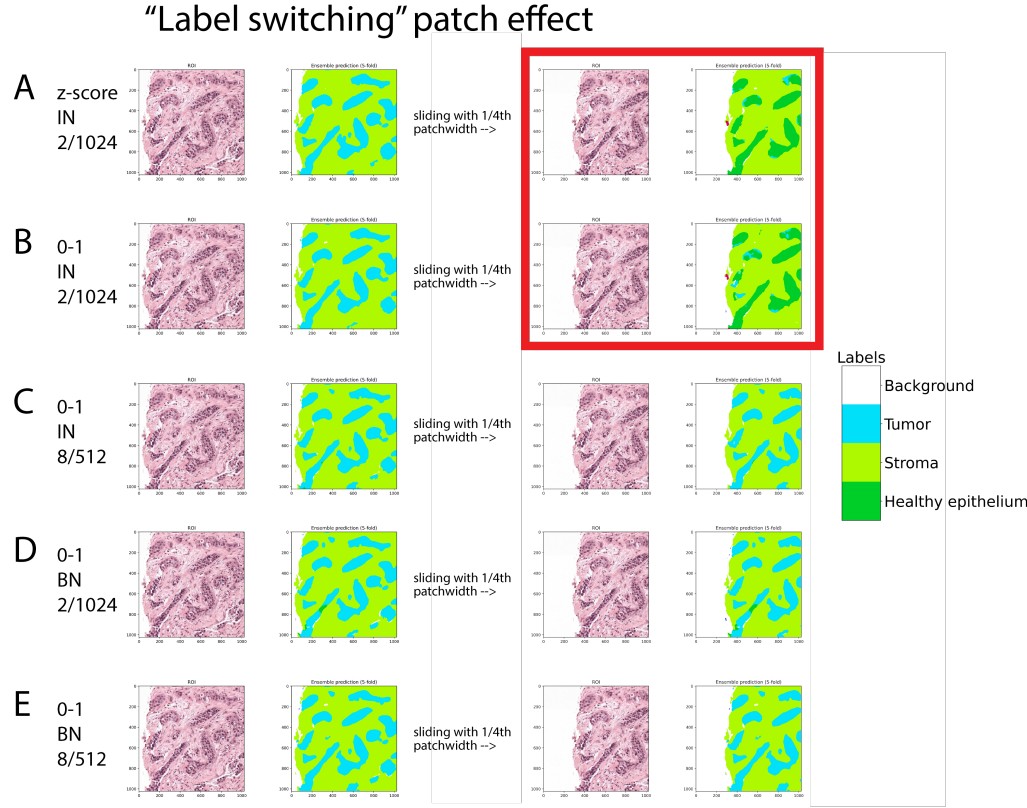

**Figure 5:** Example of the 'label-switching' problem that occurs in nnUNet's default configuration (A), and persists when only the input normalization approach is adjusted to 0-1 scaling. All experiments with a bigger batch size or BN instead of IN do not exhibit this effect (C, D, E)

## Appendix C. HoVerNet TIL detection FROC

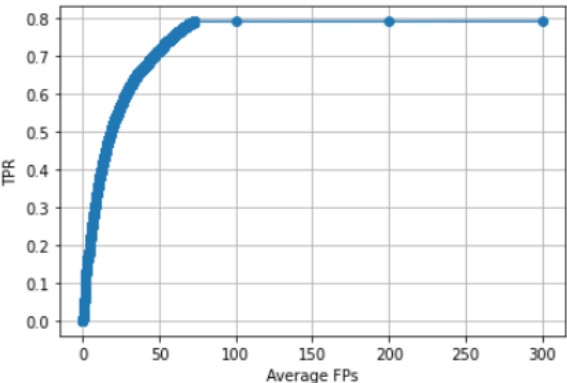

**Figure 6:** FROC analysis on HoVerNet's inflamed class as a proxy for TILs. We measure an FROC of 0.66, similar to or even surpassing TIL detections FROCs in the TIGER challenge (https://tiger.grand-challenge.org/evaluation/segmentation-and-detecton-final/leaderboard/)

## Appendix D. Inference uncertainty

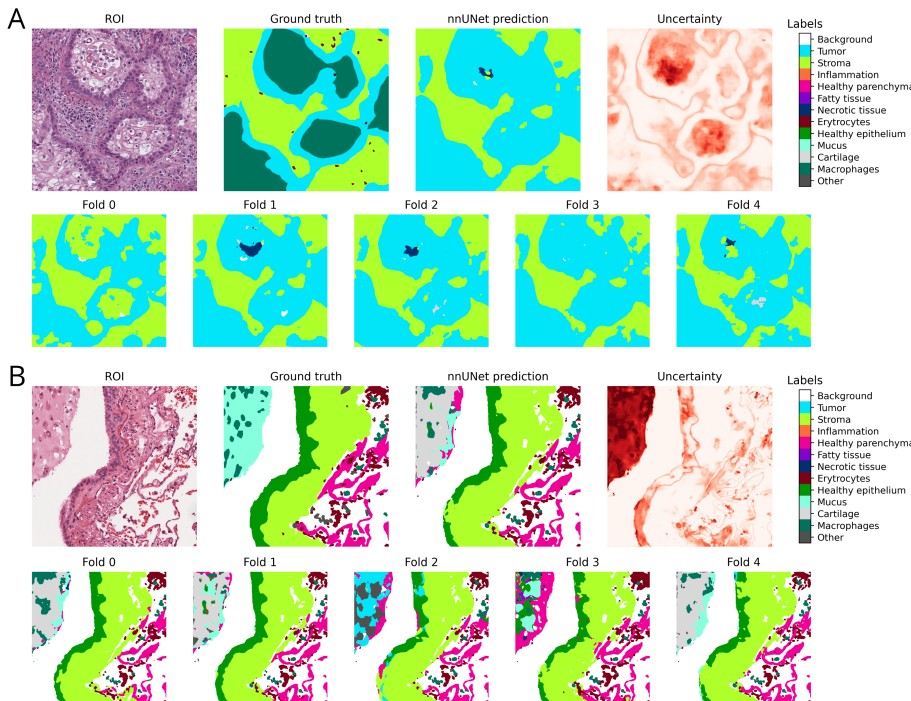

**Figure 7:** Visual examples of nnUNet's 5-fold inference output and the thereof extracted uncertainty. This image depicts uncertain regions that correlate with incorrect predictions.

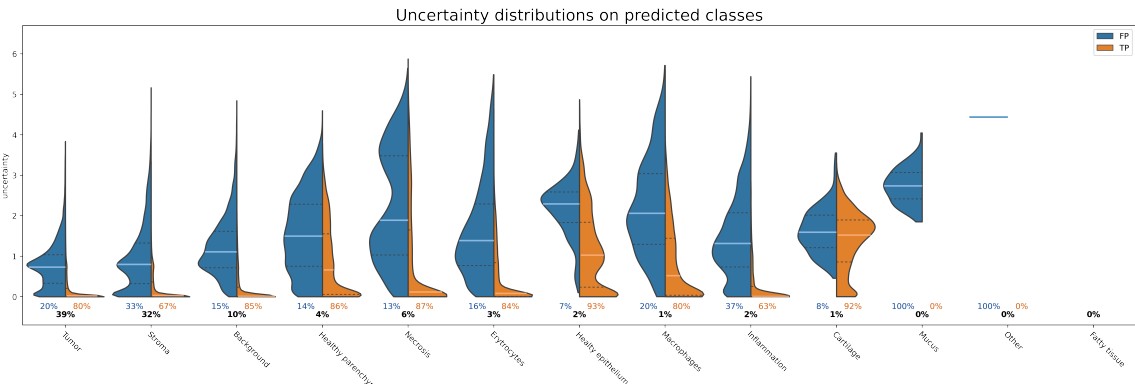

**Figure 8:** Pixel uncertainty distributions on predicted classes. (TP: orange, FP: blue). Percentages depict the sizes of the distributions.

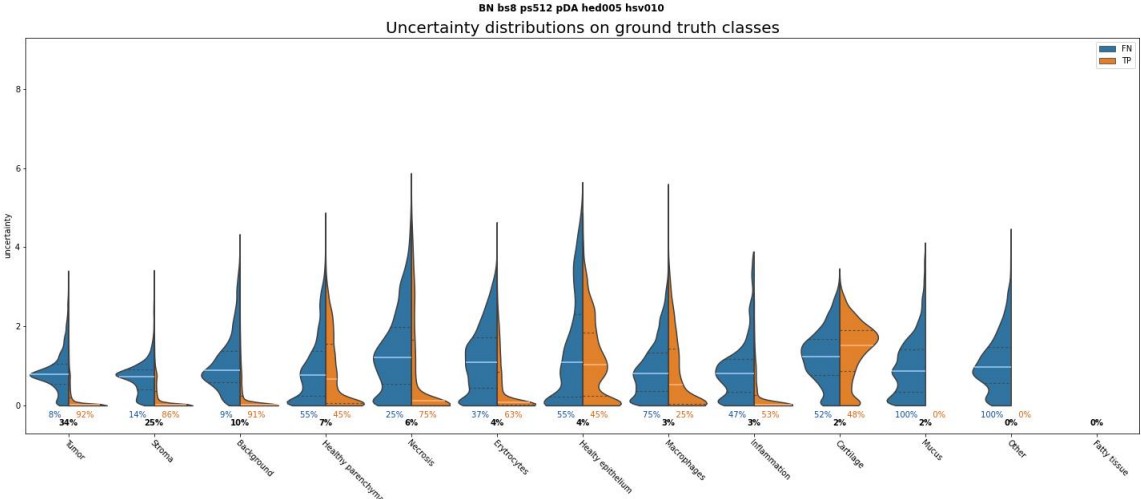

**Figure 9:** Pixel uncertainty distributions on true classes. (TP: orange, FN: blue). Percentages depict the sizes of the distributions.

## Appendix E. TIL feature Welsh tests

| TIL feature | p-value | Bonferroni corrected p-value |
|---|---|---|
| $_{tissue}$TIL | **0.040** | 0.201 |
| $_s$TILs without uncertainty filtering | **0.010** | **0.050** |
| $_i$TILs without uncertainty filtering | **0.033** | 0.163 |
| $_s$TILs with uncertainty filtering | **0.002** | **0.008** |
| $_i$TILs with uncertainty filtering | **0.019** | 0.095 |

## Appendix F. Data augmentation

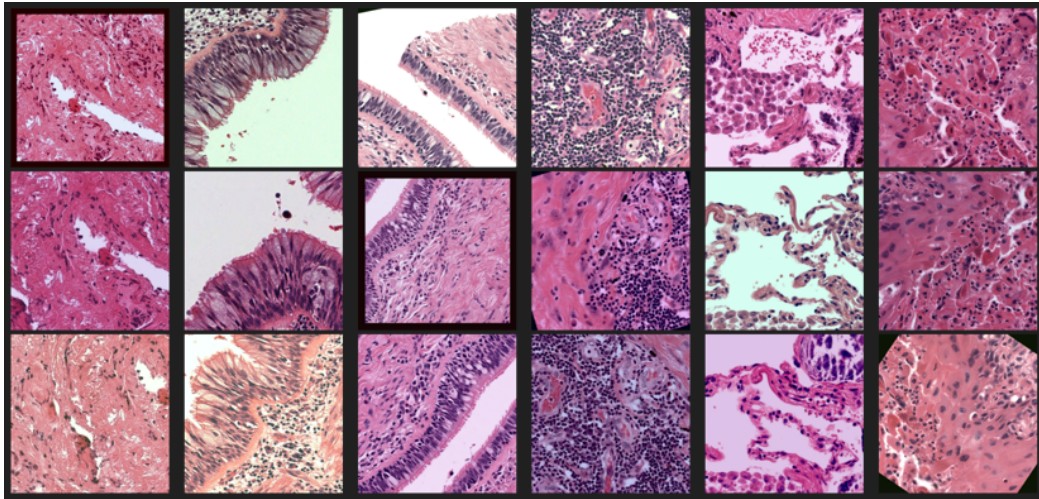

**Figure 10:** Proposed nnUNet data augmentation with additional HED light augmentations. Every column shows 3 randomly augmented versions of the same sampled patch

