# OpenReview forum: "nnUNet meets pathology: bridging the gap for application to whole-slide images and computational biomarkers"
_MIDL.io/2023/Conference — MIDL 2023 Poster_

### Official Review · Reviewer_9PBC · 2023-02-02

**Confidence:** 4
**Preliminary Rating:** 4
**Recommendation:** Poster

**Summary:**

The paper proposes to apply popular nnUNet architecture to pathology segmentation tasks. Authors make changes to the nnUNet pipeline, which allows them to apply inference to WSI images without any artifacts. They also add task-specific augmentation to the pipeline. At last, they show that they can improve immunotherapy response prediction tasks with the help of extracted segmentation uncertainties.

**Strengths:**

* Introduction provides good background on nnUNet and their shortcoming for pathology segmentation tasks.
* Good motivation about the changes in the nnUNet pipeline backed by experimental results.
* Use of segmentation uncertainties-based filtering for improving downstream immunopathology response prediction task seems novel.

**Weaknesses:**

* While I appreciate the detailed introduction about nnUNet, I think the authors may want to include more literature review of pathology segmentation work done in the field. Some relevant references [1][2]
* It would be nice if authors could have done experiments on publicly available datasets like the TIGER challenge [3] as it would allow the community to see the proposed modification's usefulness compared to other related work. The TIGER challenge dataset is also larger compared to the currently employed dataset.
* In Figure 3, the authors only show histograms for TP and FP. Why was it not shown for FN, as it is equally important compared to FPs?
* A comment about why the histogram for both TP and FP overlap for the Cartilage label would be appreciated.
* Authors mention that they select an optimal split point to discriminate likely-FP and likely-TP pixels. Is this threshold selected for each class individually, or is it a global threshold? Based on this threshold, it would be good to report the percentage of excluded TPs and FPs.
* Authors mention that excluding FP regions of the tumor and stromal segments helped. Why were FPs from only these two regions removed? Were they selected based on domain-specific knowledge, or the authors did some ablation study?
* In the Discussion section, it is mentioned that the method was evaluated on two different datasets (TCGA LUAD and TCGA LUSC), which have data from a variety of medical centers and different staining procedures. It would be interesting to see the performance difference between these two centers.
* Figure 8 in the appendix requires a better caption. Currently, it is not clear what each row or column means.
* Will the annotated private dataset made publicly available?

[1] Srinidhi, C.L., Ciga, O. and Martel, A.L., 2021. Deep neural network models for computational histopathology: A survey. Medical Image Analysis, 67, p.101813.

[2] Wang, S., Yang, D.M., Rong, R., Zhan, X. and Xiao, G., 2019. Pathology image analysis using segmentation deep learning algorithms. The American journal of pathology, 189(9), pp.1686-1698.

[3] https://tiger.grand-challenge.org/Data/

**Deanonymize Review:**

no

**Detailed Comments:**

* I think, there is a typo in Figure 4. Authors use t_TIL, it think it should i_TIL.


**Paper Type:**

validation/application paper

**Questions To Address In The Rebuttal:**

Mainly all the points raised in the Weakness section. If the authors can incorporate some of the suggestions made in the "Detailed Comments" section, it would help to improve the clarity of the paper.

Edit: After the response by the authors during the rebuttal period, I am happy to change my score to weak Accept.

---

### Official Review · Reviewer_Cftu · 2023-02-05

**Confidence:** 4
**Preliminary Rating:** 3
**Recommendation:** Poster

**Summary:**

This paper explores the use of nnU-net for pathology applications. Application that authors choose for is TIL quantification task for immunotherapy response prediction in NSCLC biopsies. There seems some uncertainty embedded into segmentation, this may improve the results.
Code seems to be available after publication.


**Strengths:**

-- application is interesting, and important
-- code will be available (just let authors know, it is still possible to share the code via anonymous links prepared for double blind reviews)
-- uncertainty quantification part is important for refining segmentations to a better stage.

**Weaknesses:**

-- there is no innovation in the paper
-- the use Unet and unet based other segmentation methods were used in pathology images already. The use nnUnet is therefore not quite interesting from innovation,
--what does figure 3 tell us?
-- Figure 2b has uncertainty induced or not?
--comparisons with SOTA missing

**Deanonymize Review:**

no

**Detailed Comments:**

The paper more weaknesses than it has advantages. It is simply an application paper with known architectures. Surely, it is not the first one either. Maybe the clinical application is more interesting, but not sure if it is suitable for MIDL then, maybe an application oriented clinical journal more appropriate.

**Paper Type:**

validation/application paper

**Questions To Address In The Rebuttal:**

the questions from weaknesses are enlisted here:
-- there is no innovation in the paper
-- the use Unet and unet based other segmentation methods were used in pathology images already. The use nnUnet is therefore not quite interesting from innovation,
--what does figure 3 tell us?
-- Figure 2b has uncertainty induced or not?
--comparisons with SOTA missing

---

### Official Review · Reviewer_uzMK · 2023-02-07

**Confidence:** 4
**Preliminary Rating:** 3
**Recommendation:** Oral

**Summary:**

This aim of this study is to explore nnUNet for pathology: our contributions are (1) developed additions to the framework to prepare pathology training data for nnUNet, and apply inference on WSI;  (2) a new proposed nnUNet configuration and added pathology color augmentations to improve tissue segmentation performance; (3) a lightweight uncertainty extraction approach that proves helpful in biomarker development. In addition the author validated this nnUNet configuration and showcase its application on a proof of concept TIL quantification task for immunotherapy response prediction in NSCLC biopsies.

**Strengths:**

This aim of this study is to explore nnUNet for pathology: our contributions are (1) developed additions to the framework to prepare pathology training data for nnUNet, and apply inference on WSI;  (2) a new proposed nnUNet configuration and added pathology color augmentations to improve tissue segmentation performance; (3) a lightweight uncertainty extraction approach that proves helpful in biomarker development. In addition the author validated this nnUNet configuration and showcase its application on a proof of concept TIL quantification task for immunotherapy response prediction in NSCLC biopsies.

**Weaknesses:**

The author performed ablation study on new nnUNet configurations and added pathology color augmentations to improve tissue segmentation performance, which could lead to relatively lack of technological novelty.

**Deanonymize Review:**

yes

**Paper Type:**

both

**Questions To Address In The Rebuttal:**

The author performed ablation study on new nnUNet configurations and added pathology color augmentations to improve tissue segmentation performance, which could lead to relatively lack of technological novelty.

---

### Official Review · Reviewer_NPzw · 2023-02-07

**Confidence:** 5
**Preliminary Rating:** 4
**Recommendation:** Poster

**Summary:**

The authors implement nnUNet for histological whole slide imaging  (H&E) multi-class segmentation including a inference trick for the WSI.
The results show that nnUNet on pathology data can achieve significant performance in immunotherapy response prediction in NSCLC biopsies with specific data pre-processing, data augmentation, and nnUNet network batch related setup.


**Strengths:**

A few technical details about WSI data pre-processing (i.e., how to convert the TIFF image to NiFTI format), data augmentation (on color, affine, etc.) are well explained in detail. For the experiment design, the nnUNet configuration test case scenarios are reasonable.

**Weaknesses:**

I fully acknowledge the technical gap between radiology images and WSI. However, my major concern is the motivation for the work. There are a few WSI-based segmentation networks (i.e., HoverNet, as authors cited in the manuscript. And we've seen transformer-based networks achieve decent results as well, i.e., Hatamizadeh, Ali, et al. "Swin unetr: Swin transformers for semantic segmentation of brain tumors in mri images." Brainlesion: Glioma, Multiple Sclerosis, Stroke and Traumatic Brain Injuries: 7th International Workshop, BrainLes 2021, Held in Conjunction with MICCAI 2021, Virtual Event, September 27, 2021, Revised Selected Papers, Part I. Cham: Springer International Publishing, 2022.).

In the validation section, the authors mainly focus on the performance of the nnUNet only. In other words, the lack of the baseline SOTA model makes the paper a bit less interesting.

To solve the inference tiling effect issue, using a partial shifting window step is not new. Please check de Bel, Thomas, et al. "Stain-transforming cycle-consistent generative adversarial networks for improved segmentation of renal histopathology." (2019).

**Deanonymize Review:**

no

**Detailed Comments:**

Authors could explain more the generalizability of the proposed data pre-processing technique for general deep networks, not only just focus on the nnUNet.

Please double-check and correct the format of the quote.

**Paper Type:**

validation/application paper

**Questions To Address In The Rebuttal:**

As per the comments above, please explain more about the motivation for the work. Why utilizing nnUNet is preferable over other frameworks, i.e., transformer-based network.

Do authors have any performance results between nnUNet and other baseline methods?

---

### Meta-Review · Area_Chair_sPZg · 2023-02-20

**Recommendation:** Accept (Poster)
**Confidence:** 5

**Metareview:**

This work received mixed scores during the initial review round, whose main criticisms were the lack of methodological novelty and comparison to state-of-the-art related models. After reading carefully the paper, the concerns raised by the reviewers, and the authors responses, I am inclined towards recommending its acceptance. While the technical novelty is rather limited, the application and adaptation of nnUNet to pathology images based on its weaknesses on this kind of images is somehow novel. Furthermore, the lack of direct comparisons to relevant prior work is addressed by the ranking achieved on the public benchmark TIGER.